# Cooperative agreement between countries of the North Atlantic Ocean reduces marine plastic pollution but with unequal economic benefits

Nicola J. Beaumont [1], Tobias Börger [2], James R. Clark [1], Nick Hanley [3], Robert J. Johnston [4], Keila Meginnis [5], Christopher Stapenhurst [6] & Frans P. de Vries [7] ✉

Plastic pollution in the world's oceans threatens ecosystems and biodiversity. The connected nature of the marine environment suggests that coordinated actions by countries sharing an ocean border may provide more effective pollution control than unilateral actions by any one country. However, countries often fail to cooperate, even when joint economic benefits would be higher under cooperation. Here we present a modelling framework to determine the potential economic benefits of cooperative marine plastic pollution management. The framework integrates an estimated plastic transfer matrix from a particle tracking model with game theory to derive the economic benefits of international cooperation for 16 countries bordering the North Atlantic Ocean. Subject to modelling uncertainties, a fully cooperative agreement yields aggregate annual net benefits of around $36 billion and a 64% reduction in emissions. The net benefits of cooperation persist over alternative scenarios and considering the impact of uncertainties but vary in magnitude and distribution.

Since the widespread use of plastic for manufacturing and packaging began in the mid twentieth century, levels of plastic production have risen dramatically to an estimated 390.7 million metric tonnes (Mt) in 2021[1]. Simultaneously, inadequate plastic waste disposal and processing systems, combined with slow rates of degradation, have resulted in plastic becoming a globally ubiquitous pollutant[2,3]. The presence of plastic pollution in the marine environment and its impact on marine wildlife are particular concerns[4,5].

It is estimated that between 4.8 and 12.7 Mt of plastic entered the world's oceans from land in 2010[6]. Subsequent work has focused on the dominant contribution of rivers as a transport pathway for plastic waste to the marine environment[7–9]. Although fewer estimates of marine inputs exist, fishing related items are frequently reported in marine litter surveys[10]. The flux of plastic to the oceans is predicted to increase in the future, with planned interventions unable to prevent ongoing accumulation and associated negative impacts[11,12].

A range of international frameworks include strategies to address marine plastic pollution (MPP), including the European Union (EU) Marine Strategy Framework Directive, the OSPAR Convention, the HELCOM Convention, and the Barcelona Convention. Most recently, UN Member States voted to establish an Intergovernmental Negotiating Committee (INC) with the mandate of advancing an international legally binding instrument on plastic pollution, including the marine environment[13]. As the marginal benefits and costs of reducing plastic waste vary between countries, effective and cost-efficient reductions of MPP will require international cooperation[14,15].

International cooperation to address reductions in any transboundary global or regional pollutant is challenging, because the benefits of these reductions have the characteristics of a public good[16]. The physical movement of plastic in and across international waters implies that some of the environmental damages due to plastic waste emitted from one country may be imposed on other countries. Emissions reductions by any one country (e.g., country A) also decrease marine plastics in the waters and beaches of other countries (e.g., countries, B, C and D), even if the latter countries make no efforts to reduce their own emissions. Consequently, although

[1]Plymouth Marine Laboratory, Plymouth, UK. [2]Berlin School of Economics and Law (HWR Berlin), Berlin, Germany. [3]School of Biodiversity, One Health and Veterinary Medicine, University of Glasgow, Scotland, UK. [4]George Perkins Marsh Institute, Clark University, Worcester, MA, USA. [5]Patient-Centered Research, Evidera; School of Biodiversity, One Health and Veterinary Medicine, University of Glasgow, Scotland, UK. [6]Quantitative Social and Management Sciences Research Centre (QSMS), Faculty of Economics and Social Sciences, Budapest University of Technology and Economics, Budapest, Hungary. [7]Department of Economics, Business School, University of Aberdeen, Scotland, UK. ✉e-mail: frans.devries@abdn.ac.uk

expenditures on MPP reductions from any one country are incurred by that country alone, the benefits of unilateral abatement actions are experienced by all countries whose territorial waters and shorelines are impacted. The resulting non-excludability of benefits (i.e., actions by one country necessarily benefit others) creates a strategic cooperation problem, since it can be in each country's selfish interest to rely on abatement actions undertaken by other countries with which they share the ocean resource, rather than reducing emissions themselves. Possible strategies to solve this cooperation dilemma include (i) international agreements whereby each country agrees to a legally-binding mandate to reduce its own emissions if other countries also act, and/or (ii) cooperative agreements wherein countries offer each other side payments to induce greater cooperation[14,15].

Recognizing this problem, the objective of this paper is to bring timely insights to bear on the net benefits of international cooperation over MPP reductions, and the conditions under which mutually beneficial coordination can occur. We illustrate a framework for assessing the potential economic benefits of cooperation that incorporates empirical estimates of (i) plastic transfer coefficients between countries and (ii) the economic benefits of reduced MPP for each country. We integrate these elements using an adaptation of the game-theoretic framework in Mäler[17], initially applied to the case of sulphur dioxide (SO₂) emissions that contribute to acid rain[18]. The features of the control problem which make our adaptation of Mäler's approach particularly appropriate are (i) representation of the physical realities of pollutant transfer across the North Atlantic to different countries, (ii) variation in both the valuation of damage costs across the affected countries, along with variation in the costs of reducing pollution, (iii) an optimisation framework in which the benefits of cooperation can be quantified relative to a baseline of no cooperation, and (iv) each country's incentive to behave non-cooperatively in terms of abatement expenditures. Results from the integrated assessment provide insight into key questions for international coordination, including: (1) What abatement policy maximises the net economic benefits of international cooperation in MPP reduction? (2) How are the benefits of international cooperation distributed between different countries under the optimal cooperative solution? (3) What is the impact of political-economy constraints on these cooperative outcomes, in terms of both the overall benefits of cooperation and the reduction in MPP?

Whilst the approach presented here shares a similar conceptual foundation to Mäler's[17] framework, the differences between MPP and SO₂ pollution lead to important variations in model structure. A key difference between MPP and SO₂ relates to the dynamics of MPP movement through the natural environment. Unlike SO₂, MPP can visit and cause damage in multiple locations before settling in a fixed location or being broken down. Further, our knowledge of plastic emissions to the ocean, and the transport dynamics and fate of different types of plastic while in the ocean, carry large uncertainties[19–22]. Finally, the spatial distribution of MPP and countries' geographical location has implications for their strategic position regarding the incentive to reduce transboundary pollution in a multi-regional setting[23,24]. These factors affect the economic benefits of international cooperation over MPP reductions, the likelihood of such cooperation, and what can be done to encourage it.

We use this framework to investigate the benefits of international cooperation over MPP for countries bordering the North Atlantic Ocean. Because the ocean transport and environmental impacts of MPP vary over different types of plastic (e.g., micro- versus macroplastic)[4], the information required to implement the model, and the potential answers to our research questions, also differ. We thus limit the scope of the study by focussing on floating macroplastic (> 0.5 cm in size) that has entered the marine environment via rivers, before being moved within and between each country's territorial waters. This decision makes the model consistent with the accompanying empirical data collected on the economic benefits of MPP reduction. It also reflects the availability of plastic emissions data[7,8], and the current state of knowledge regarding the distribution of plastic in the global ocean, which tends to be better for surface plastics which are easier to observe and sample[25].

In this framework, transfer coefficients describe the accumulation of MPP in each country's territorial waters given an estimate of their respective plastic emissions to the marine environment and the movement of plastic throughout the ocean. We use a country's Exclusive Economic Zone (EEZ) to define its territorial waters (Fig. 1a). Given MPP can sequentially visit the waters of multiple countries during its time in the ocean, we define the transfer coefficients in terms of the contribution of different countries to the annual mean stock of marine plastic in each country's EEZ. This contrasts the work of Mäler[17] on SO₂, where annual depositional fluxes were used to generate the transfer matrix.

The MPP transfer coefficients were calculated using a particle tracking model[26] with gridded surface ocean currents and wind data[27,28] which was constructed specifically for this project; and annual estimates of river plastic emissions[8]. We simplify the model by fixing annual emissions. This step simplifies the economic modelling by providing a set of transfer coefficients that are in near steady state, subject to remaining interannual variations due to variable ocean and wind forcing. A parameterisation of plastic removal was used to account for the selective loss of buoyant plastic from surface waters to the subsurface and seafloor[10], and to bring the model to an approximate steady state. This was calibrated to align estimates of plastic river inflows with the generally smaller estimates of surface ocean plastic inventories that have been derived from observational and model-led studies[20,29]. The idealised setup accounts for different plastic residence times within EEZs and solves the problem of lag effects associated with varying plastic transit times between countries, with the caveat that time-varying inputs are not accounted for. The final model yields a matrix of transfer coefficients, where each value gives the annual average fractional contribution of each country's emissions to the stock of plastic in each country's EEZ (Fig. 1). Details about the model assumptions, its calibration and estimation of these figures are given in Methods and in the Supplementary Information (SI).

The modelled countries in the North Atlantic include Belgium (BE), Canada (CA), Denmark (DK), Dominican Republic (DO), France (FR), Germany (DE), Haiti (HT), Ireland (IE), Mexico (MX), Morocco (MA), Netherlands (NL), Portugal (PT), Spain (ES), Sweden (SE), United Kingdom (UK), and United States (US), as explained in the SI. The pathways connecting countries are illustrated using a network diagram (Fig. 2), which shows the fraction of the surface plastic stock exported to adjoining waters per day, averaged over the final year of the model simulation.

A set of marginal damage cost estimates per unit of plastic emissions in each country's national waters and coastline was obtained based on original stated-preference estimates for the US and the UK[30], and converted for each of the other 14 nations using a unit-value benefit transfer[31], with adjustments for income and purchasing-power parity (PPP). A country's marginal benefit from reducing its plastic emissions by one unit—equivalent to the marginal damage cost per unit of unabated emissions—is equal to the economic value of the ecosystem damages in its own coastal waters and beaches that are avoided due to this reduction in plastic emissions. Theoretically and empirically, this may be quantified using estimates of each country's willingness to pay (WTP) for MPP reductions. Unique benefit estimates are provided for MPP reductions that impact the beaches and coastal waters of each modelled country.

The economic optimisation model then uses these two sets of parameters (plastic transfer coefficients and countries' WTP for MPP reductions) to calculate each country's non-cooperative (selfish) level of plastic emissions into the ocean, defined as the point at which its domestic marginal abatement costs are equal to the marginal benefit of emission reductions in that country. These benefits pertain only to MPP reductions on countries' own beaches and coastal waters. This yields a unique, non-cooperative Nash Equilibrium (NE) level of MPP abatement for each country, reflecting a situation from which no individual country has an incentive to deviate, given that it will not improve their individual payoff. The same model is used to simulate the vector of abatement levels for each country which, instead, maximises net benefits (value of damage reductions minus abatement costs) across all 16 countries combined, considering the benefit of each country's

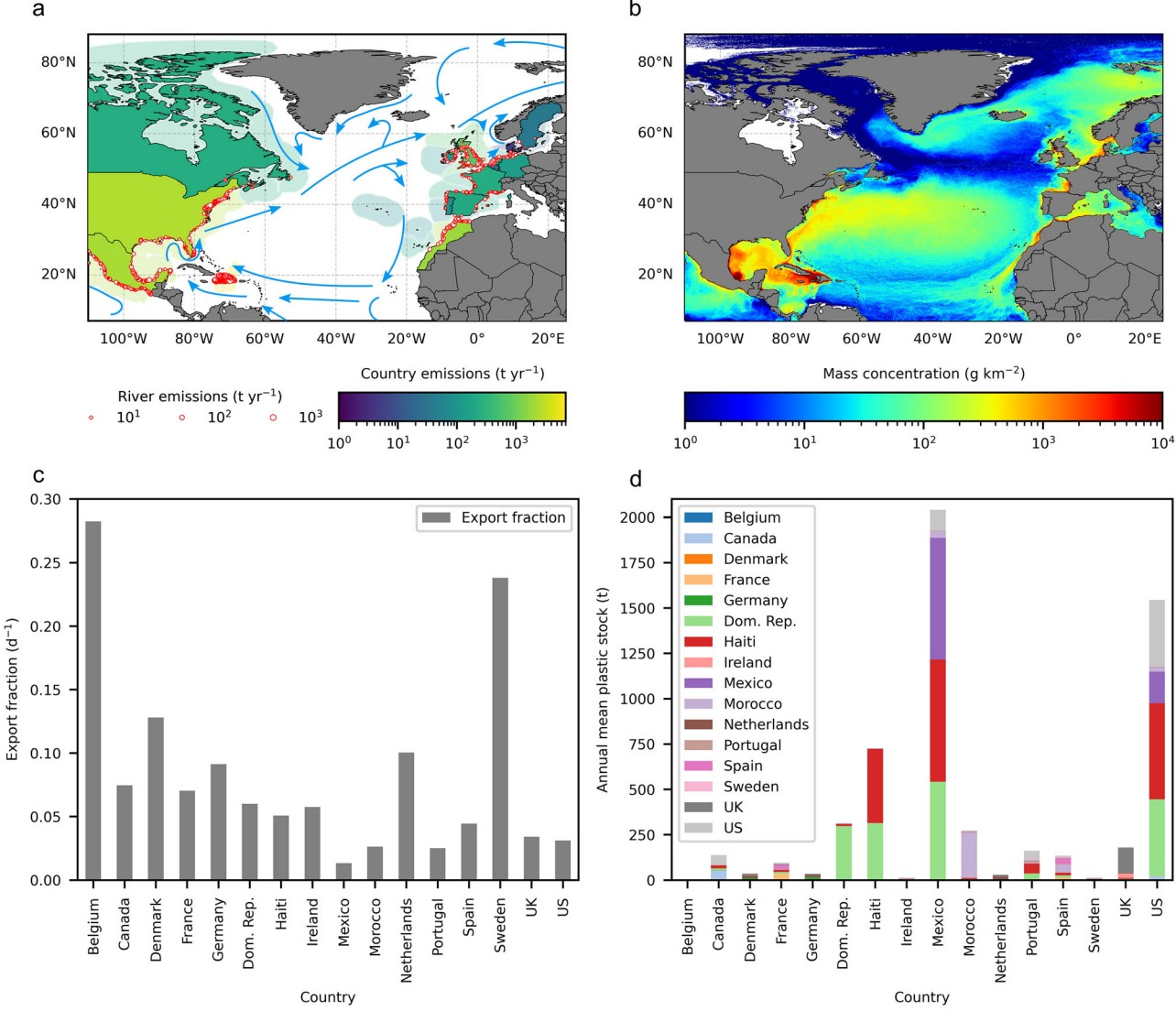

**Fig. 1 | The area covered in the study and summary results from the plastic transfer model. a** Map showing the countries included in the study; the extent of their respective EEZs; their river emissions; and the major ocean currents influencing the transfer dynamics of surface ocean plastic in the region. **b** Map showing the modelled, annual mean mass concentration of surface ocean macroplastic in the North Atlantic in the year 2014. **c** The annual mean of surface ocean plastic within each EEZ which is exported per day to the EEZ of a neighbouring country, or to other waters, in the year 2014. **d** The annual mean stock of surface ocean plastic within the EEZ of each country in the year 2014, with colours indicating the country from which the plastic originated.

emission reductions on all other countries. This forms the optimal "full cooperation" scenario, which can be compared to the non-cooperative NE, both for each country individually and when summed across all 16 North Atlantic countries, while accounting for uncertainties in the underlying transfer coefficients and marginal damage cost estimates (see Methods). This full cooperation outcome characterises the distribution of plastic abatement which maximises net benefits summed across all 16 countries. This is the optimal, cooperative MPP abatement outcome from a coalition of cooperating nations who all agree to become members of a binding International Environmental Agreement (IEA).

## Results

### Plastic transport modelling

While the plastic transport model is idealised, it produces spatial patterns in the distribution of plastic that are consistent with past observational and modelling studies, including the accumulation of plastic in the North Atlantic Subtropical gyre[32–34]. In the final year of the simulation, the annual mean mass of MPP at the surface of the North Atlantic Ocean and surrounding regional seas is 9.0 thousand metric tonnes, which is of the same

order as past estimates generated using three separate models that had been calibrated against data for plastics measuring <200 mm in size[20]. The dominant contribution of large plastics to the total plastic budget has been reported[29,35], although estimated inventories vary considerably; and there remain relatively few robust observations of surface plastics >200 mm in size to validate models against, especially in the North Atlantic Ocean[29]. Ultimately, as the economic model normalises the transfer coefficients, the absolute value for the stock is not used. It is the relative contribution of each country to the stock in a given EEZ that is of critical importance, with country differences determined by plastic transport dynamics and the parameterisation of loss and decay.

The sensitivity to the type of buoyant plastic is explored by reducing the wind drift factor (Supplementary Fig. 5). In general, lower wind drift factors are representative of less buoyant objects that sit lower in the water. Using a metric for exposure (see SI), a wind drift factor of 1% yields a percentage change in exposure that can exceed 100% relative to runs using the default value of 2% (Supplementary Figs. 5a, b). However, the highest differences are generally limited to countries that are geographically separated and thus only weakly connected, meaning their impact on the overall transition

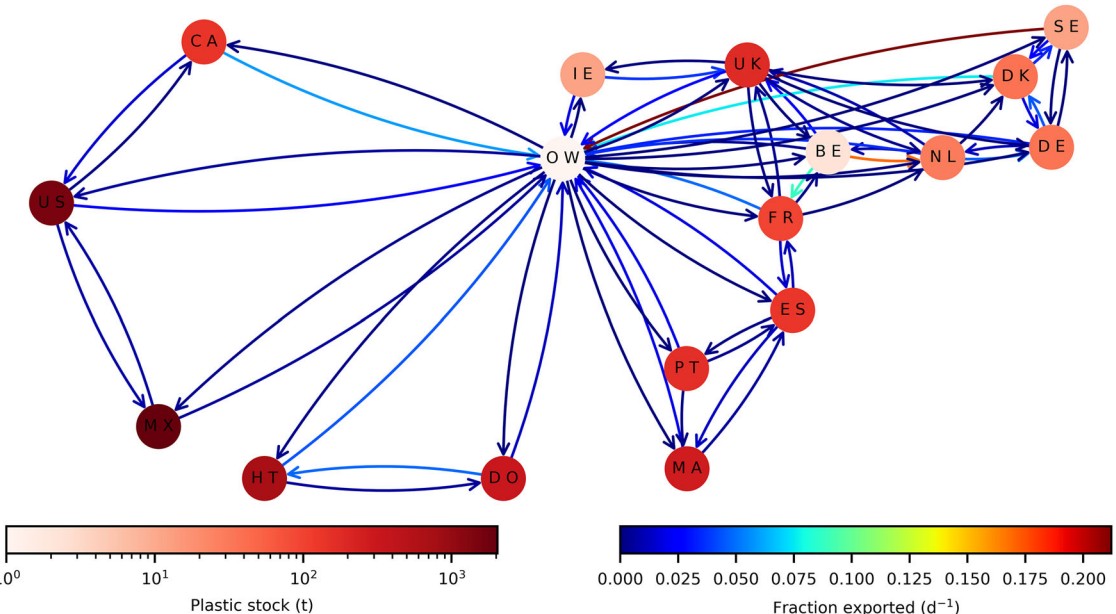

**Fig. 2 | Network diagram showing the stocks and flows of plastic within and between the EEZs of countries included in the study.** Other Waters (OW), representing international waters and the territorial waters of all countries not included in the study, are included to highlight their important role in mediating transfers between geographically separated territories. The relatively large stock for OWs has been omitted from the diagram to help highlight differences in the stock in each country's EEZ.

matrix is small. As expected, differences become more marked when the effect of the wind is excluded (Supplementary Figs. 5c, d).

Asymmetries in the exchange of MPP between neighbouring countries are evident in Fig. 2 and follow known features of the ocean circulation in these areas. These include the residual eastward flow of water through the English Channel into the Southern North Sea and eventually up along the coast of Norway via the Norwegian Coastal Current; and the north eastward movement of water from Haiti and the Dominican Republic toward the Gulf of Mexico and the coasts of Mexico and the US. Although recent studies have used labels and other features of plastic items to try and determine their origin[36,37], at present there is a lack of comprehensive data on the origins of recovered MPP and the relative contribution of countries to each other's stock of plastic pollution. Therefore, it is not possible to validate the idealised stock estimates shown in Fig. 1. We explore the impact of uncertainty in these estimates on the economic modelling to help quantify the model's robustness below.

## Economic modelling

We take the sum of the non-cooperative NE to yield zero additional abatement over current (status quo) emission levels. Countries are assumed to start from this outcome, and all results are then obtained relative to this equilibrium of national-level marginal benefits and costs. In contrast to this case where each country maximises its own net benefits unilaterally and independently from other countries' abatement decisions, we assume the purpose of an IEA is that signatory countries choose an abatement policy that increases (and ideally maximises) collective net benefits. This means that each country reduces their plastic emissions up to the point where their marginal cost of abatement is equal to the sum of all countries' marginal benefits. Based on transfer coefficients for the final year of a 15-year simulation that started in 2000 and ended in 2014, this economically efficient abatement policy results in a reduction of 64% in plastic emissions to the North Atlantic (Fig. 3d), compared to the non-cooperative NE outcome. The net collective benefits equal around 62% of the total damage caused by MPP to the sample countries (Fig. 3b). However, these benefits are shared unevenly across countries, with the US and Germany receiving a higher share of the

benefits (Fig. 3b). Countries tend to benefit more if they have high national income (since this drives the benefit estimate adjustment for each country) and/or if they receive a large share of the plastic emissions from other countries.

The largest share of abatement activity in the cooperative solution is allocated to countries who have the lowest marginal abatement costs and/or are responsible for more MPP transferred beyond their EEZ. This is explained by the fact that countries who transfer a higher fraction of their plastic pollution (i.e., countries with a relatively high transfer coefficient) have less incentive to abate in the absence of an IEA, hence are required to do relatively more due to the agreement. Countries whose plastic emissions are transferred to countries with higher gross national income (GNI) implement more abatement in the optimal cooperative solution than countries whose plastic emissions are transferred to lower GNI countries. For example, Mexico has higher abatement costs than Canada; yet it is required to conduct more than three times as much abatement in the economically efficient outcome as Canada because it has a larger impact on the stock of plastics in the US, which is the highest GNI country and thus has a relatively high WTP for abatement. Some countries, such as Dominican Republic and the Netherlands, are made individually worse off as a result of this cooperative outcome.

## Impact of political-economy constraints

Since the gains from cooperation and abatement responsibility are unevenly shared in the efficient solution, we explored alternative scenarios in which the cooperative solution is subject to a range of political-economy constraints. The potential constraints are: (C1) no country is allowed to increase their plastic emissions relative to the baseline ("positive abatement" scenario); (C2) every country must derive equal net benefit from the agreement ("equal value" scenario); (C3) every country must agree to reduce plastic emissions by the same percentage relative to the baseline ("equal abatement" scenario); (C4) no country is allowed to experience negative benefits from the agreement ("positive value" scenario).

The outcomes of these four scenarios are compared to the economically efficient cooperative outcome (Fig. 4 and Supplementary Table 3). When no country is allowed to increase emissions (C1) relative to the

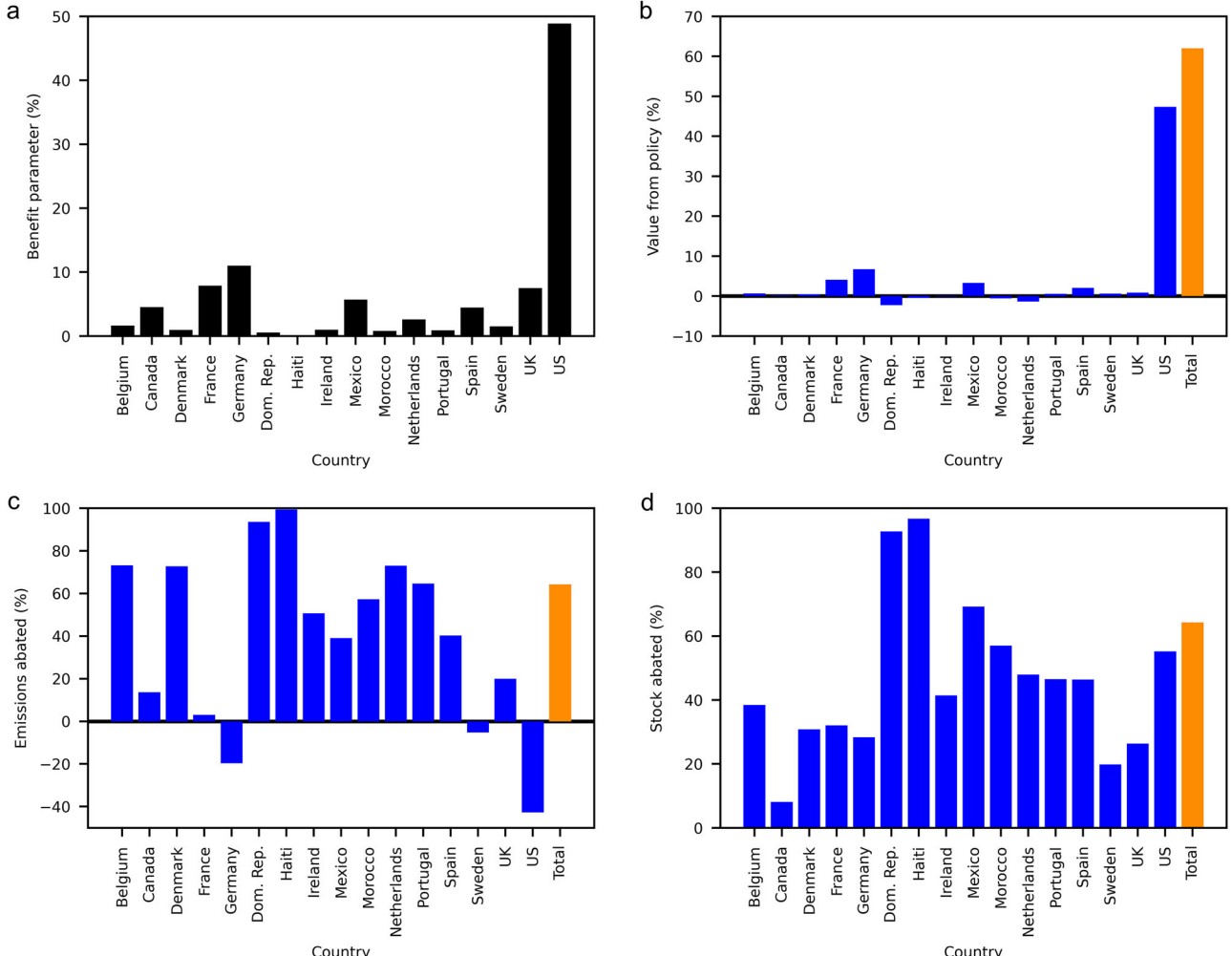

**Fig. 3 | Results from the economic optimisation model for the fully cooperative scenario, evaluated in the year 2014. a** The benefit parameter which has been normalised to allow it to be expressed in percentage terms. Its conversion to US dollars is accomplished by multiplying through by twice the WTP to abate all pollution ($59bn) and dividing through by 100. **b** The economic value that each country gains from the policy expressed as a percentage of the total WTP. The vertical orange bar shows the total value to all countries. **c** The percentage of each country's river emissions that are abated. The black horizontal line marks zero abatement. The vertical orange bar shows the total percentage of river emissions abatement when summed across all countries. **d** The percentage of the total plastic stock abated. The vertical orange bar shows the total percentage of all plastic stock abated when summed across all countries. Associated quantitative data is shown in Supplementary Table 2.

baseline, there is a small reduction in the overall economic net benefits, and a small increase in total plastic abatement from 64% to 67% of current emissions. The equal value scenario (C2) leads to large losses of potential aggregate benefits from international cooperation. When each country is required to cut emissions by an equal amount (C3), total plastic abatement falls from 62% to 43% of the baseline, and much lower net benefits are realised. When the policy is constrained to ensure that all countries are better off (C4), the net benefits and the percentage of MPP abatement fall by more than half, from 62% to 27% and from 64% to 25%, respectively. Thus, the least damaging constraint in terms of both overall MPP reductions and net economic benefits is that of ensuring no country is allowed to increase its baseline emissions (C1).

### Robustness and sensitivity
A key question for this integrated model is how uncertainty surrounding the estimated transfer coefficients affects the gains from cooperation. Compared to constraints C2 and C3, the relative dominating performance of the IEA under constraint C1 in terms of MPP reductions and the derived economic benefits is robust to small changes in the transfer coefficient estimates (Supplementary Tables 4 and 5). The second most predominant channel affecting potential changes in the benefits from cooperation is determined by countries' relative GNI position and differences in (marginal) abatement costs[38]. Sensitivity analyses carried out with respect to the estimated income elasticities reveals that greater inequality in these elasticities increases the value of the agreement. This is because a country's share of the overall benefit from MPP abatement is increasing with their income elasticity (see Methods). Moreover, we also find that the gains from cooperation increase when a country's abatement costs and benefit from abatement are positively correlated, and vice versa[39]. This is driven by the assumption that the status quo abatement choices constitute a non-cooperative outcome and are determined by how much a country's MPP stays within domestic borders, i.e., reducing MPP increases benefits. Finally, the results are robust in terms of countries' relative cost-benefit position, depending on whether a country faces high/low benefit and/or high/low cost. In particular, the gains from cooperation between high benefit-high cost (HiHi) countries and low benefit-low cost (LoLo) countries are determined by the fraction of MPP that the HiHi country receives from the LoLo country, whilst the fraction of MPP in the LoLo country coming from the HiHi country is of little consequence[38] (see Methods for a detailed sensitivity analysis).

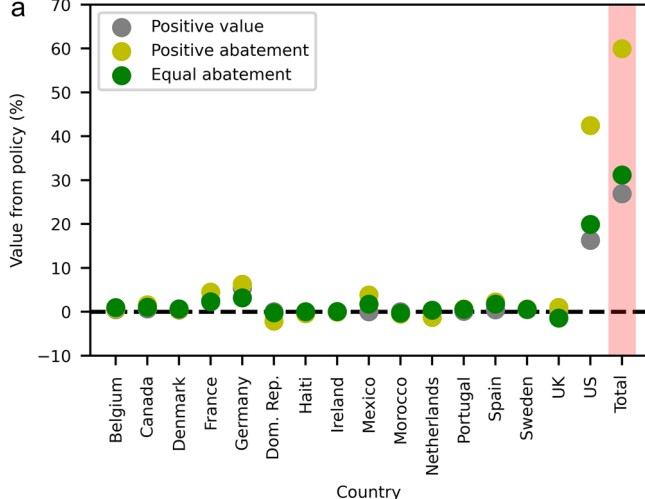

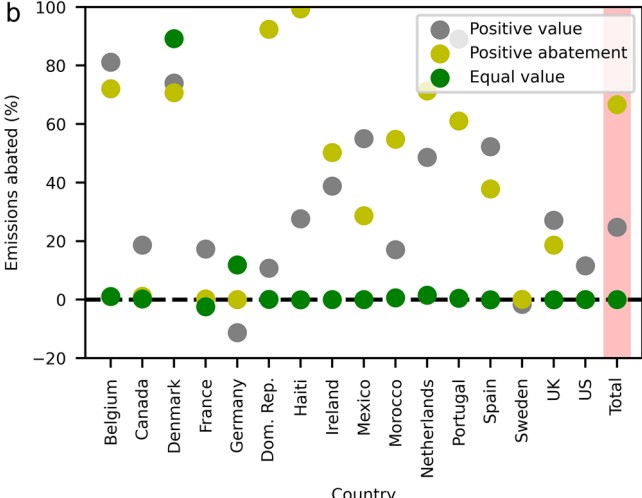

**Fig. 4 | Results from the economic optimisation model for the year 2014 under political-economy constraints C1–C4 (see main text for details). a** Value of the policy in percentage terms. In the case of equal value constraint (C2), the value was 0% for all countries. The vertical red line is used to highlight the total value summed across all countries. **b** Emissions abated in percentage terms. In the case of equal abatement constraint (C3), abatement was 43% for all countries. The vertical red line is used to highlight the total emissions abated summed across all countries. Associated quantitative data is shown in Supplementary Table 3.

## Discussion

Our research develops a generalisable framework for analysing the potential gains from international cooperation when reducing MPP. The analysis demonstrates how ocean modelling can be linked to economic valuation and game-theoretic analysis to assess the benefits of cooperation, and the effects of alternative political-economy considerations on the IEA performance. Results were drawn from a particular case study—buoyant macroplastic pollution in the North Atlantic. At present, detailed knowledge regarding key model components remains uncertain, including knowledge regarding the movement of plastic between different reservoirs as well the benefits and costs of MPP abatement beyond the scenarios included in our stated preference study, which includes buoyant macroplastic only, and only samples residents in two of the North Atlantic countries (the US and UK). The analysis was able to assess the robustness of the findings regarding a number of assumptions. However, future studies could explore potential extensions of the approach to incorporate features such as a more comprehensive representation of how plastic moves from sources to sinks[35] (while accounting for associated uncertainties), time-varying conditions

(e.g., MPP inputs; changes in GNI), or potential feedbacks between the introduction of abatement targets and enhanced abatement technologies. For example, we assume that MPP has reached a steady-state equilibrium in the physical environment. But MPP has been steadily increasing over time, is forecast to continue increasing, and moves slowly in the ocean—features that reduce the likelihood of a steady state. Moreover, if MPP is increasing over time, gains from international cooperation are likely to be higher than those reported here. Such effects could be explored using an Integrated Assessment Model, building on the framework developed here.

Further steps to improve the presented framework could include incorporation of (i) additional abatement benefits, (ii) different types of plastic, (iii) a more expansive set of countries, and (iv) original data on abatement costs. For example, our results may understate the full economic benefits of reducing MPP, since the valuation data used to estimate the benefits of reduced MPP for each country relate only to the values citizens attach to reducing pollution in their own national beaches and coastal waters[30]. But citizens may also value MPP reductions in international waters and in third-party countries, implying that the benefits of abatement could be higher. Future work could expand the framework to incorporate broader representation of the environmental costs of MPP and hence the benefits of abatement[40]. Moreover, as emphasized above, the approach (and WTP estimates in Börger et al.[30]) only consider reductions in buoyant macroplastic, whereas an IEA could also reduce stocks of other types of plastic. We also exclude plastic entering the North Atlantic from countries outside of our sample. These limitations cause us to overestimate the total fraction of plastic abated by the optimal policy, and hence the potential value of the optimal policy. Parallel approaches could be developed to incorporate other types of MPP, including microplastics, and additional countries beyond our North Atlantic sample.

We also assume that countries have chosen historic MPP emissions optimally given their individual costs and benefits. However, understanding of MPP damages continues to evolve, suggesting that past emissions may not have been chosen with full information on costs and benefits. Recent unilateral policy changes in some countries (e.g., single-use plastic bans) further suggest that actual abatement costs may be less than those we infer from each country's non-cooperative Nash equilibrium abatement level. To address these limitations, future work would benefit from the inclusion of original data on country-specific abatement costs.

Assumptions and limitations such as these are common in large-scale integrated models and unavoidable in some form to maintain tractability. Caveats such as these notwithstanding, the presented framework demonstrates the existence of potentially large collective gains from international cooperation over reductions in MPP in the North Atlantic. The structure of the approach, along with our sensitivity analyses, suggest that the benefits of cooperation are likely to persist even with adaptations of the framework to address limitations such as those described above. The potential benefits of cooperation are particularly important given on-going international efforts to secure a global agreement on MPP. In our model, the optimal "full cooperation" agreement for the North Atlantic case yields aggregate net benefits across all 16 countries of around $36 billion per annum, and results in an overall reduction of 64% in baseline emissions.

Importantly, however, countries benefit unequally (and some lose) from the economically efficient outcome, with large abatement burdens falling on a few low-income countries. As shown in our robustness and sensitivity analysis, constraining the solution to bar countries from increasing emissions relative to the baseline is relatively low cost and achieves similar levels of total abatement, but does not ensure that all countries benefit. In contrast, an equal burden sharing agreement, wherein each country agrees to the same percentage cut in plastic emissions, forgoes many of the benefits of full cooperation, as does an agreement constrained to ensure that each individual country gains more from the global policy than the non-cooperative equilibrium. These losses in aggregate net benefits relative to the efficient outcome reflect the asymmetry in transfer coefficients and marginal damage costs across countries. When countries lose from a potential IEA, the incentives to

participate are reduced or eliminated, unless a credible side-payment scheme can be put in place whereby countries who gain agree to compensate the countries who lose[41]. More robust agreements ensure that no country (or coalition) can gain by deviating from the agreement and free-riding on the increased abatement efforts of others[42].

## Methods

Empirical data is used to calibrate a particle tracking model to compute transfer coefficients. Benefit estimates are derived from a random utility, WTP space model estimated using data from a previously published stated-preference choice experiment implemented in the US and the UK[30]. These benefit estimates are then extended to all 16 North Atlantic countries using simple benefit transfer methods based on differences in PPP-adjusted national income. Finally, (economic) optimisation techniques are used to compute (i) each country's selfish, non-cooperative equilibrium, used here as a baseline; (ii) the abatement levels and net benefits arising in a full cooperative agreement which maximises aggregate net benefits summed across all 16 countries, and (iii) abatement levels and net benefits in a range of other potential cooperation agreements, referred to as "political-economy constraints."

### Plastic transfer model

The transboundary nature of the problem is captured by an $N \times N$ transfer matrix $T$. The $(i,j)$th entry of the transfer matrix denotes the fraction of plastic originating from country $i$ that contributes to the stock in country $j$'s waters and coasts. We used the particle tracking model PyLag[26] to simulate the movement of plastic between emitting and receiving countries. The PyLag model is fully open source and includes a suite of online examples (https://pylag.readthedocs.io/en/latest/) that demonstrate the model's ability to reproduce the results of analytical models, and to function with data defined on different grids. The model was configured to read in driving data from separate Ocean and Atmospheric General Circulation Models, which it used to compute simulated particle trajectories and associated connectivity metrics.

In the study, we focussed on the surface transfer of buoyant plastic only and did not attempt to simulate the movement of subsurface plastic or associated subsurface and seafloor inventories. The choice to focus on the surface ocean only is consistent with its important role in mediating the transfer of plastic between locations[21]; and with the areas referred to in questions used to determine respondents' maximum WTP for plastic abatement, from which benefit estimates were derived[30]. We used surface ocean currents and a simple model of leeway[43], which combines the effects of windage and Stoke's Drift, to simulate the movement of buoyant plastic in the ocean. To facilitate the use of the model outputs in the economic modelling, we configured it to asymptote toward a steady-state solution for the total inventory of buoyant plastic at the ocean surface. This was achieved by assuming a constant rate of emissions; and by allowing particle weights—representing the mass of plastic each particle represents—to decay in time. Particle decay coefficients were pulled from a uniform distribution, with lower values allowing plastic particles to travel longer distances before their mass at the ocean surface is reduced to a negligible level (Supplementary Fig. 1). In this way, the model was parameterised to produce a surface ocean inventory and global distribution of floating plastic debris which is consistent with existing published estimates[20,29].

For a simulated particle on the sea surface with position vector $X = X(r, t)$, where $r = X(t = t_0)$ is the particle's initial position vector, trajectories were computed using a Stochastic Differential Equation of the form:

$$dX_i = \left[ u_i + \frac{\partial A_h}{\partial x_i} \right] dt + \left( 2A_h \right)^{1/2} dW_i, \qquad (1)$$

where $dX_i = dX$ is the incremental change in particle's position in the interval $[t, t + dt]$; $u_i$ is a velocity term; $A_h$ is the isotropic, horizontal eddy diffusivity; and $dW_i$ is an incremental Wiener process that builds

stochasticity into the model. As we only consider motion at the ocean surface, vertical advection and diffusion are ignored.

The velocity, $u(t)$, is:

$$u = u_o + \alpha u_w, \qquad (2)$$

where $u_o$ is the Eulerian surface ocean velocity at the location $x(t) = X(r, t)$, $u_w$ is the corresponding surface wind velocity at a height of 10 m above the ocean surface, and $\alpha$ is the leeway or wind factor, which determines an object's sensitivity to wind forcing. Here, we make the simplifying assumption that wind forcing drives the particle in a direction parallel to the downwind direction. This assumption can be contrasted with search and rescue models, which often attempt to account for leeway divergence using deflection angles derived from the tracks of objects at sea. Deflection angles typically fall in the range of −30° to +30° and depend on the type of object being modelled[43]. For the wind factor, we used a constant value of 0.02. The value is based on PyLag modelling of GPS tracked plastic bottles (unpublished data), which represents one of the few datasets on how a common type of MPP moves along the ocean's surface. Although we adopt a fixed value for this parameter, it is expected to vary between litter types. The sensitivity of the model results to this parameter is explored in Supplementary Fig. 5.

The impact of unresolved motions on particle movement are included through $A_h(x, t)$. $A_h$ is derived from the surface ocean velocity field using the Smagorinsky equation[44]:

$$A_h = \frac{1}{2} C \Delta^2 \left( \left( \frac{\partial u_o}{\partial x} \right)^2 + \left( \frac{\partial v_o}{\partial y} \right)^2 + \frac{1}{2} \left( \frac{\partial u_o}{\partial y} + \frac{\partial v_o}{\partial x} \right)^2 \right)^{1/2}, \qquad (3)$$

where $C$ is the Smagorinsky constant, which is set equal to 0.2; and $\Delta^2$ is taken to be the area of the element in which the particle is located.

We used daily surface ocean current data (for variable $u_o$) covering the period 1995–2014, which were taken from the 1/12° CMEMS Global Ocean Physics Analysis GLORYS12V1 dataset[27]; and hourly surface winds (for variable $u_w$) covering the same period from the approximately 1/4° ERA5 dataset[28]. Using the particle tracking model PyLag[26], both grids were decomposed into a set of spherical triangular elements covering the Earth's surface, with velocity components defined at element nodes. Interpolation of velocity components within elements was performed using each particle's spherical barycentric coordinates, giving $C^0$ continuity. Gradients in the velocity field within elements, which were required for calculations of $A_h$, were computed by first rotating the corresponding Cartesian axes so the positive $z$-axis forms an outward normal through the element's centroid, and the $x$- and $y$-axis is locally aligned with lines of constant longitude and latitude, respectively. The element is then projected onto the plane that lies tangential to the surface of the sphere at the element's centroid, and gradients in $x$ and $y$ calculated[45].

The model was integrated forward in time by applying separate discrete integral operators for advection and diffusion. For advection, a fourth-order Runge-Kutta scheme was used with a time step of 3600 s. A Milstein scheme is used for the stochastic diffusive component, using a time step of 900 s.

In the triangular grid, all land elements were masked. If a particle transitioned into a masked element during the integration, it was reflected back into the ocean. Thus, the transfer model does not explicitly resolve beach inventories; rather, these are subsumed within the in-water inventory and are treated as one when used with the economic model. This is a pragmatic choice, which is motivated by the current uncertainty associated with modelling beaching explicitly using coarse global ocean data[34].

### Plastic decay model

Several studies have noted an apparent mismatch between estimates of plastic emissions into the ocean, and estimates of surface ocean plastic inventories, as derived from in situ observations and model simulations[20,29]. These studies suggest the inventory of floating plastic litter is 1–2 orders of

magnitude less than estimated land-based emissions of plastic to the ocean each year. Many studies have since explored mechanisms to explain the apparent discrepancy. These include, but are not limited to, accumulation in rivers[46], beaching[47] and coastal trapping[48]. Another recent study concluded the flux from rivers is lower that estimated[35], which helps to close the budget. However, the model includes several processes that are difficult to parameterise (e.g., beaching), and the relative importance of these factors and mechanisms has not been conclusively established.

Our plastic decay model is motivated by the finding that different types of non-buoyant and buoyant plastic dominate in the coastal and open ocean, and in surface and seafloor environments[10]; and the requirement to simulate steady-state conditions for use with the economic model. To allow the system to approach a near steady-state inventory, each particle was associated with a weighting factor, $w$, with units of tonnes. Weighting factors were allowed to decay as a function of time, $t$. The mass of particle $i$ emitted from river $j$ is then given by:

$$w_{ij}(t) = \frac{E_j}{N_P} e^{-\lambda_i t},$$ (4)

where $E_j$ is the mass of plastic emitted by river $j$ each month; $N_P$ (= 100) is the number of particles released from each river; and $\lambda_i$, with units of inverse time, is the decay factor for particle $i$. Each particle is associated with a decay constant from the set $\{0.1, 0.2, 0.3, \ldots, 9.9, 10.0\}$, corresponding to a set of e-folding time scales that range from just over a month up to 10 years.

The model is an idealisation and designed to simulate a steady state. However, at the same time it is designed to reproduce observed patterns that suggest some types of buoyant plastic rapidly collect on the sea floor, while others stay at the ocean's surface for longer and undergo far-field transport; e.g., into the sub-tropical gyres, where plastics have been observed to be decades old. Supplementary Fig. 1 shows how this model allows Belgium's emissions to transition toward a steady-state inventory over the course of the simulation. Further details on how particle weighting factors are used to compute plastic stocks and flows can be found in the SI. The sensitivity of plastic stock and flow estimates to the choice of evaluation year and the value of the wind factor are shown in Supplementary Figs. 2–5.

### Economic model

We adopt the transboundary pollution model of Mäler[17]. In this model, countries choose how much pollution abatement to carry out given the national-level costs and benefits of abatement. The gains from cooperation are given by the difference between a country's payoffs when they internalise the impact of their abatement choices on other countries versus when they ignore them. We formalise and discuss each of these components in turn. The model was implemented and coded in Mathematica (version 12.2).

There is a set of countries $N$, with a generic country labelled by $i \in N$. Each country $i$ chooses to abate a percentage $a_i \leq 1$ of its current ("status quo") plastic emissions. If $a_i = 0$, then country $i$ continues to emit its status quo level of polluting emissions; if $a_i < 0$, then country $i$ increases its level of emissions relative to the status quo. We denote the vector of abatement choices by $a := (a_i)_{i \in N}$ (we use notation ":=" to indicate that the symbol on the left is defined by the expression on the right).

The cost of abatement choice $a_i$ is described by a country-specific cost function, $c_i(a_i) := -\gamma_i \ln(1 - a_i)$, where $\gamma_i \geq 0$ is country $i$'s cost parameter. This function has the desirable properties that (i) the cost of maintaining the status quo level of abatement is normalized to 0 (i.e., $c_i(0) = 0$); (ii) the marginal cost of abatement at the status quo level is exactly equal to cost parameter (i.e., $c_i'(0) = \gamma_i$); (iii) the cost and marginal cost are increasing in the fraction of MPP abated (i.e., $c_i' > 0$ and $c_i'' > 0$); (iv) the marginal cost of abating MPP is infinite at full abatement (i.e., $\lim_{a_i \to 1} c_i'(a_i) = \infty$).

The transboundary nature of the problem is captured by an $N \times N$ backward transition matrix, $S$ (Fig. 1d). The $(i, j)$th entry of this matrix denotes the fraction of the stock of buoyant plastic in country $j$'s waters that originated from country $i$. We calculate it from the transfer matrix $T$ by

dividing each $(i, j)$th entry by the sum of the entries in column $j$ to give $S_{ij} = \frac{T_{ij}}{\sum_{i \in N} T_{ij}}$. If countries commit to a vector of abatement choices, $a$, then the stock of plastic in country $j$'s EEZ falls by a fraction $R_j(a) := \sum_i a_i S_{ij}$. We refer to this as country $j$'s "received abatement." The vector of received abatement is given by $R(a) := \left(R_j(a)\right)_{j \in N} = a \cdot S$. The total stock of plastic reduces by $\frac{\sum_{j \in N} \sum_{i \in N} a_i T_{ij}}{\sum_{j \in N} \sum_{i \in N} T_{ij}}$. The transport model assumption that MPP has reached a steady state implies that the transfer matrix is constant across time.

All national-level economic benefits to country $i$ from abatement relate to damage reductions in the coastal waters and beaches of that country. This presumes that each country receives no benefit from reductions in the stock of MPP outside that country's EEZ. Country $i$'s benefit of a percent of abatement $R_i$ is described by a country-specific quadratic[49,50] benefit function, $b_i(R_i) = \beta_i R_i(2 - R_i)$, where $\beta_i \in \mathbb{R}$ is a country-specific damage parameter. Note that $b_i(1) = \beta_i$ and $b_i'(0) = 2\beta_i$, so $\beta_i$ is both country $i$'s benefit from full abatement and half its marginal benefit of abatement at the status quo level. The other two quadratic parameters are chosen to normalize the benefits of the status quo emissions to zero ($b_i(0) = 0$), and to ensure that marginal benefits are continuous at full abatement ($b_i'(1) = 0$).

Country $i$'s normalized payoff ("value") from abatement vector $a$ is the difference between its costs and benefits, divided by the global benefit of abating all MPP, $B := \sum_{i \in N} \beta_i$:

$$v_i(a) := \frac{b_i(R_i(a)) - c_i(a_i)}{B}.$$ (5)

The outcome of abatement vector $a$ is the vector of resulting values, $v(a) := (v_i(a))_{i \in N}$, and the total value is the sum $V(a) := \sum_{i \in N}(a)$. If country $i$ non-cooperatively maximises its own value, then it chooses $a_i$ to satisfy the first-order condition (FOC) $\frac{\partial v_i(a)}{\partial a_i} = 0$. In our context, country $i$'s FOC can be written as:

$$\frac{\gamma_i}{1 - a_i} = 2\beta_i(1 - R_i(a))S_{ii},$$ (6)

which indicates that the marginal benefit of a unit of pollution abatement to country $i$ is equal to the marginal cost of abatement, given that it can only affect a fraction $S_{ii}$ of its incoming plastic pollution. A strategy profile $a^*$ is a pure-strategy NE if Eq. (6) holds for all countries $i \in N$. The corresponding Nash value vector is $v(a^*)$, and the Nash value is $V(a^*)$. Existence of this equilibrium is guaranteed[51] if we make the reasonable assumption that there is a lower bound on how much pollution each country can abate (or equivalently, an upper bound on how much pollution each country can emit).

Following Mäler[17], we assume that the status quo abatement choices are part of a non-cooperative NE. This means that Eq. (6) is satisfied for all countries for $a^* = 0$. This identifies country $i$'s cost parameter as:

$$\gamma_i = 2\beta_i S_{ii}.$$ (7)

Mäler[17] uses this condition to identify the unknown damage (benefit) parameters, $\beta_i$, from his empirical marginal abatement cost estimates, $\gamma_i$. We do the reverse: we use empirical estimates of $\beta_i$ (discussed below) to identify $\gamma_i$.

The cooperative abatement profile, $a^{**}$, is that which maximises aggregate net benefits over all countries combined, $V(a)$. The fact that $b_i$ is concave in $R_i$ and $c_i$ convex in $a_i$ implies that each $v_i$ is concave in $a$. This subsequently implies that $V(a)$ is concave in $a$ (and strictly concave so long as at least one of the benefit or cost parameters is non-zero). Therefore, any solution $a^{**}$ to the system of FOCs, $\frac{\partial V(a^{**})}{\partial a_i} = 0 \ \forall i \in N$, characterises the

unique global maximum. In our case, this condition simplifies to:

$$\frac{y_i}{1-a_i^{**}} = 2\sum_{j \in N} \beta_j \left(1 - R_j(a^{**})\right) S_{ji} \qquad \forall i \in N. \tag{8}$$

This indicates that each country $i$ should abate at the level that sets their marginal cost of reducing domestic emissions (the left-hand side of (8)) equal to the marginal benefits of this reduction in pollution to all countries impacted by country $i$ (the right-hand side of (8)). Substituting Eq. (7) into (8) yields:

$$\frac{\beta_i}{1-a_i^{**}} S_{ii} = \sum_{j \in N} \beta_j \left(1 - R_j(a^{**})\right) S_{ji} \qquad \forall i \in N. \tag{9}$$

The strategy profile $a^{**}$ is the optimal cooperative strategy profile. Equation (9) reveals that this strategy depends on neither the levels of the damage parameters $\beta_i$, nor on the levels of the plastic quantities $T_{ij}$, but only on their relative values. The cooperative value vector is $v^{**} := v(a^{**})$, and the cooperative value is $V^{**} := V(a^{**})$.

For a given NE strategy $a^*$, the gains from cooperation are equal to the difference $V^{**} - V(a^*)$. Our assumption that status quo abatement is a NE implies $V(a^*) = 0$, and thus the gains from cooperation are equal to exactly $V^{**}$. We necessarily have $V^{**} \geq V(a)$ for all strategies $a$, including any NE $a^*$, because $a^{**}$ is the unique maximiser of $V^{**}$. It follows that the gains from cooperation are always positive. This does not imply that $v_i(a^{**}) < v_i(a_i^*)$ for each individual country $i$, so it may be the case that some countries are worse off in the cooperative outcome.

## Benefit parameter estimates

We obtain estimates of the benefit parameters $\beta_i$ for the UK and the US from a discrete choice experiment (DCE) designed to provide estimates of per household WTP for different quantities and types of MPP abatement. Methods and data from these DCEs are described by Börger et al.[30]. We use a mixed logit model[52,53] in WTP space of the benefit functions $b_{UK}$ and estimate that the average UK household is willing to pay \$1.510 to abate 1% of buoyant plastic from UK domestic beaches, and \$1.559 to abate 1% of buoyant plastic from the UK coastal waters. Models were run in $R$[54] using the 'Apollo' package[55,56] to produce these estimates (details in SI). Summing these provides a lower-bound household WTP of \$3.069 for abating 1% of MPP from the UK's EEZ. With 28.535 million households in the UK[57], this national WTP estimate is approximately \$88mn. It follows that $b_{UK}(0.01) = \beta_{UK}0.01\left(1 - 0.01/2\right) = \beta_{UK}0.00995 \approx \$88mn$, such that $\beta_{UK} \approx 88mn/0.00995 \approx 8.8bn$. Similarly, for the US we obtain WTP estimates of \$2.252 and \$2.196 for abating 1% of beach and coastal pollution, respectively, yielding a total of \$4.448 to abate 1% of buoyant plastic from US waters. There are roughly 128.45 million households in the US[58], and following the same routine yields $\beta_{US} = 57bn$.

Employing international benefit transfer methods[59], we use income- and purchasing power parity (PPP)-adjusted unit-value transfers to estimate WTP for the other countries in the set. This is required because primary data required to estimate these values is only available for the US and UK. If the WTP of a household $i$ is known, a common unit-value benefit transfer technique estimates the WTP of another household $j$ using the relationship $\text{WTP}_j = \left(\frac{y_j}{y_i}\right)^\epsilon \text{WTP}_i$, where $y_i$ and $y_j$ are the households' respective incomes, and $\epsilon$ is an estimate of the income elasticity parameter[59]. Since our primary concern is on countries rather than households, we adapt this method by estimating country $j$'s WTP parameter as $\beta_j = \left(\frac{y_j}{y_i}\right)^{\epsilon_{GNI}} \beta_i$, where $y_i$ and $y_j$ are the GNI of countries $i$ and $j$, respectively, and $\epsilon_{GNI}$ is the elasticity of the national WTP parameter with respect to GNI. Summing over $j$ and using the normalization $\sum_{j \in N} \beta_j = 1$ reveals that $\beta_i = \frac{y_i^{\epsilon_{GNI}}}{\sum_{j \in N} y_j^{\epsilon_{GNI}}}$. This approach is used because it obviates the need to obtain and reconcile estimates of household numbers across all countries in the simulation, as

needed for traditional benefit aggregation and benefit transfer. It should be noted this applied approach will only yield equivalent results to those of a standard income- and PPP-adjusted unit-value benefit transfer[59] for the restricted case in which $\epsilon_{GNI} = \epsilon = 1$. For all other cases, the GNI-adjusted benefit transfer will produce comparatively larger benefit estimates for countries with a larger number of households when $\epsilon > 1$, and smaller estimates when $\epsilon < 1$.

The fact that we already have WTP estimates for the UK and the US means that we can use the equality $\beta_{UK} = \left(\frac{y_{UK}}{y_{US}}\right)^{\epsilon_{GNI}} \beta_{US}$ to calibrate the income elasticity parameter as $\epsilon_{GNI} = \frac{\log(\beta_{UK}) - \log(\beta_{US})}{\log(y_{UK}) - \log(y_{US})}$. Substituting the $\beta_{UK}$ and $\beta_{US}$ estimates into this expression, together with purchasing power parity (PPP) adjusted 2021 GNI estimates[60] of \$3327bn and \$23,393bn for the UK and US, respectively, yields an estimate of $\epsilon_{GNI} = 0.96$, which is commensurate with measures found in other studies[59].

## Sensitivity analysis

This section quantifies the sensitivity of our results to our estimates of the model parameters $T$, $\epsilon$, and $y := (y_i)_{i \in N}$. For these purposes, it is useful to define $\theta := (T, \epsilon, y)$ and to write $V(a; \theta)$ instead of the shorthand $V(a)$. The total gains from cooperation can be interpreted as the value of an optimisation problem, $\mathcal{V}(\theta) := \max_a V(a; \theta)$, with optimal policy $a^{**} = argmax_a V(a; \theta)$. The envelope theorem[61] tells us that $\partial \mathcal{V}(\theta)/\partial \theta = \partial V(a; \theta)/\partial \theta|_{a=a^{**}}$. Letting $I$ denote the indicator function, we obtain (see SI):

$$\partial \mathcal{V}/\partial T_{ij} = 2\beta_j \left[\left(a_i - R_j(a^{**})\right)\left(1 - R_j(a^{**})\right)\right.$$
$$\left. + \left(I(i=j) - S_{jj}\right)\ln\left(1 - a_j^{**}\right)\right] / \sum_{k \in N} T_{ki}$$
$$\partial \mathcal{V}/\partial \epsilon = \epsilon \sum_{i \in N} \ln(y_i)\left(v_i(a)/V - \beta_i\right) \tag{10}$$
$$\partial \mathcal{V}/\partial y_j = \epsilon\left(v_j^{**} - V^{**}\beta_j\right)/y_j.$$

The elasticity of the total gains from cooperation with respect to the quantity of MPP transitioning from country $i$ to country $j$ is then:

$$\frac{\partial \mathcal{V}}{\partial T_{ij}} \frac{T_{ij}}{V^{**}} = 2\beta_j S_{ij} \left[\left(a_i^{**} - R_j(a^{**})\right)\left(1 - R_j(a^{**})\right)\right.$$
$$\left. + \left(I(i=j) - S_{jj}\right)\ln\left(1 - a_j^{**}\right)\right]/V; \tag{11}$$

the elasticity with respect to the income of country $i$

$$\frac{\partial \mathcal{V}}{\partial y_i} \frac{y_i}{V} = \epsilon\left(v_i^{**}/V^{**} - \beta_i\right); \tag{12}$$

and the elasticity with respect to the income weight parameter $\epsilon$ equal to

$$\frac{\partial \mathcal{V}}{\partial \epsilon} \frac{\epsilon}{V} = \epsilon \sum_{i \in N} \ln(y_i)\left(v_i^{**}/V^{**} - \beta_i\right). \tag{13}$$

## Data availability

Driving data for the particle tracking model, including data on river plastic emissions, are available from the public sources listed in the GitHub repository https://github.com/pmlmodelling/beaumont_et_al_plastics_econ. The derived transfer matrices have been deposited in the same GitHub repository and can be accessed at https://github.com/pmlmodelling/beaumont_et_al_plastics_econ/tree/main/plastic_transfer/Results/plastic_stocks. Other outputs from the particle tracking model, including particle positions as a function of time, are available from the author James Clark upon request. The authors declare that all data relating to the economic

**Article**

modelling are available within the paper and the Supplementary Information file.

## Code availability

The particle tracking model PyLag v0.7 is open-source software (https://github.com/pmlmodelling/pylag/). The online documentation contains tutorials on how to use the model to perform particle tracking simulations (https://pylag.readthedocs.io/en/latest/). Configuration and analysis code for both the plastic transfer and economic models can be accessed at https://github.com/pmlmodelling/beaumont_et_al_plastics_econ.

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

## Acknowledgements

This work was conducted as part of the project "The Economics of Marine Plastic Pollution—What are the Benefits of International Cooperation?" funded by the UK Economic and Social Research Council (ES/S002448/2). The support of the ESRC is gratefully acknowledged. This work used the ARCHER2 UK National Supercomputing Service (https://www.archer2.ac.uk). Stapenhurst gratefully acknowledges funding by the Sustainable Development and Technologies National Programme of the Hungarian Academy of Sciences (FFT NP FTA). The findings, interpretations and conclusions presented are entirely those of the authors, and any errors remain those of the authors alone.

## Author contributions

Idea conceptualisation: N.H., F.d.V.; Obtaining original research funding: N.B., T.B., J.C., N.H., R.J., F.d.V.; Design of transfer model simulations: J.C., C.S., F.d.V., N.H.; Estimation transfer coefficients: JC; Choice experiment design and analyses: T.B., N.H., R.J., K.M.; Economic modelling: C.S., F.d.V.; Graphics: J.C.; Contribution to writing: All authors.

## Competing interests

The authors declare no competing interests.
