## [Transparent Peer Review file · Communications Earth & Environment]

Cooperative agreement between countries of the North Atlantic Ocean reduces marine plastic pollution but with unequal economic benefits

Corresponding Author: Professor Frans de Vries

Version 0:

Decision Letter:

Dear Professor de Vries,

First of all, please allow me to apologise for the delay in sending a decision on your manuscript titled "Assessing Economic Benefits of International Cooperation over Marine Plastic Pollution". It has now been seen by 2 reviewers, and we include their comments at the end of this message. They find your work of interest, but some important points are raised. We are interested in the possibility of publishing your study in Communications Earth & Environment but would like to consider your responses to these concerns and assess a revised manuscript before we make a final decision on publication.

We therefore invite you to revise and resubmit your manuscript, along with a point-by-point response that takes into account the points raised. Please highlight all changes in the manuscript text file.

In particular, for publication in Communications Earth & Environment, we request that you

- (1) Provide comprehensive methodological details and justify your methodological choices over the other approaches.
- (2) Consider expanding your analysis for different types of plastics and changing trends over time and discuss the limitations of your approach and findings.

Please submit your point-by-point responses as a separate file, distinct from your cover letter where you can add responses to the Editors' comments that you do not want to be made available to the reviewers. Word files are preferred. We recommend that any figures, tables or graphs that are included in the response to reviewers are also included in the main article or Supplementary Information.

Please use the following link to submit your revised manuscript, point-by-point response to the referees' comments (which should be in a separate document to any cover letter), a tracked-changes version of the manuscript (as a PDF file) and the completed checklist:

Link Redacted

We hope to receive your revised paper within six weeks; please let us know if you aren't able to submit it within this time so that we can discuss how best to proceed. If we don't hear from you, and the revision process takes significantly longer, we may close your file. In this event, we will still be happy to reconsider your paper at a later date, as long as nothing similar has been accepted for publication at Communications Earth & Environment or published elsewhere in the meantime.

Please do not hesitate to contact us if you have any questions or would like to discuss these revisions further. We look forward to seeing the revised manuscript and thank you for the opportunity to review your work.

Best regards,

Yuwan Duan
Editorial Board Member
Communications Earth & Environment
orcid.org/0000-0002-0557-7525

Martina Grecequet, PhD
Senior Editor
Communications Earth & Environment

EDITORIAL POLICIES AND FORMATTING

Editorial Policy: [Policy requirements](https://www.nature.com/documents/nr-editorial-policy-checklist.pdf) (Download the link to your computer as a PDF.)

- Behavioural and social science
- Ecological, evolutionary & environmental sciences
- Life sciences

<https://www.nature.com/documents/nr-reporting-summary.zip>

Furthermore, please align your manuscript with our format requirements, which are summarized on the following checklist: [Communications Earth & Environment formatting checklist](https://www.nature.com/documents/commsj-phys-style-formatting-checklist-article.pdf)

and also in our style and formatting guide [Communications Earth & Environment formatting guide](https://www.nature.com/documents/commsj-phys-style-formatting-guide-accept.pdf) .

*** DATA: Communications Earth & Environment endorses the principles of the Enabling FAIR data project (<http://www.copdess.org/enabling-fair-data-project/>). We ask authors to make the data that support their conclusions available in permanent, publically accessible data repositories. (Please contact the editor if you are unable to make your data available).

All Communications Earth & Environment manuscripts must include a section titled "Data Availability" at the end of the Methods section or main text (if no Methods). More information on this policy, is available at <http://www.nature.com/authors/policies/data/data-availability-statements-data-citations.pdf>.

If a community resource is unavailable, data can be submitted to generalist repositories such as [figshare](https://figshare.com/) or [Dryad Digital Repository](http://datadryad.org/). Please provide a unique identifier for the data (for example a DOI or a permanent URL) in the data availability statement, if possible. If the repository does not provide identifiers, we encourage authors to supply the search terms that will return the data. For data that have been obtained from publically available sources, please provide a URL and the specific data product name in the data availability statement. Data with a DOI should be further cited in the methods reference section.

Please refer to our data policies at http://www.nature.com/sdata/policies

<http://www.nature.com/authors/policies/availability.html>

REVIEWER COMMENTS:

Reviewer #1 (Remarks to the Author):

This study assesses the economic benefits of international cooperation in addressing marine plastic pollution, a critical global issue. I hope my comments will contribute to the improvement of this valuable research.

The primary objective of this study is to demonstrate that international cooperation is essential for tackling the problem of plastic pollution. However, I believe the study should also address why international cooperation is often challenging. What are the key factors that hinder effective international collaboration on this issue? Additionally, what strategies can be employed to overcome these challenges?

The authors estimate the plastic transfer matrix using a game theory model. I find it difficult to understand how such a transfer matrix can be accurately estimated based on a game-theoretical approach. The movement and distribution of plastic pollution are influenced by numerous physical factors. How does the study justify estimating this matrix using an economic model rather than a model grounded in physical science?

The study relies heavily on Malei's work, highlighting the differences between Marine Plastic Pollution (MPP) and sulfur dioxide (SO₂) pollution. However, are there other models more suitable for analyzing MPP, given its unique characteristics and the high degree of uncertainty associated with it? The authors should clarify why Malei's model was chosen and consider whether alternative models might be more appropriate for this study.

Figure 4: The authors mention that the scenario of equal value is not depicted. However, it appears that an equal value scenario is present in Figure 4b. Could the authors please clarify this discrepancy?

Reviewer #2 (Remarks to the Author):

Reviewer:

Assessing Economic Benefits of International Cooperation over Marine Plastic Pollution

Summary of the Paper:

The paper estimates the economic consequences of international cooperation in controlling marine plastic pollution (MPP) by analyzing the experiences of 16 North Atlantic countries. It develops a framework combining plastic transfer coefficients with a game-theoretic model to evaluate the potential economic gains of cooperation. The paper highlights how coordinated efforts can lead to significant MPP reductions and increased economic benefits, contrasting fully cooperative and non-cooperative scenarios. It explores political economy considerations, which are crucial for achieving equitable outcomes among nations and understanding the constraints that influence cooperation.

Strengths of the Paper:

1. Integration of oceanographic modeling with Game Theory:

The paper stands out by merging oceanographic data on plastic transfer dynamics with game-theoretic models. This combination enhances the understanding of how marine plastic pollution crosses national boundaries and provides a quantitative basis for analyzing economic efficiency in international cooperation. The use of plastic transfer coefficients is a novel approach that incorporates environmental factors into the economic analysis, making the results more relevant for policymakers.

2. In-depth analysis of cooperation scenarios:

By comparing non-cooperation with full cooperation, the paper offers a thorough and well-structured economic analysis. It provides insights into how varying levels of international cooperation impact pollution reduction and welfare outcomes. The addition of political economy constraints, such as equal abatement and equal benefits, deepens the analysis by reflecting real-world complexities in international agreements.

3. Policy-relevant findings:

The paper's findings have practical relevance for international policymaking. It quantifies the economic benefits of cooperation on MPP, offering policymakers evidence to support the formation of cooperative agreements. It emphasizes the need for region-specific solutions and demonstrates that substantial gains can be achieved through joint action, which is particularly relevant for ongoing negotiations under the UN framework for a global plastic treaty.

Key Claims:

1. Economic benefits of cooperation:

The authors convincingly argue that international cooperation leads to significant economic benefits, backed by a model that integrates plastic transfer dynamics with willingness-to-pay data. The application of game theory to MPP is innovative and provides valuable insights into cooperative strategies.

2. Unequal distribution of benefits:

The paper highlights that the benefits of cooperation are unevenly distributed, with wealthier nations, such as the United States and Germany, gaining more from MPP reduction efforts while large burdens of plastic abatement falling on a few, low-income countries. This conclusion is supported by economic modeling that considers income disparities, plastic emissions, and transfer coefficients.

Suggestions for improvement:

1. Incorporate more empirical data on the costs of MPP in sectors like tourism and fisheries to broaden the context.
2. Expand the analysis to include different types of plastics, beyond macroplastics.
3. Add dynamic elements to the model to account for changing plastic emission trends over time.

Conclusion:

The paper makes a valuable contribution to environmental economics by applying game theory to international cooperation on marine pollution. Limitations of the study should be clarified. Nevertheless, the work provides a strong foundation for understanding the economic benefits of cooperative MPP management.

Communications Earth & Environment is committed to improving transparency in authorship. As part of our efforts in this direction, we are now requesting that all authors identified as 'corresponding author' create and link their Open Researcher and Contributor Identifier (ORCID) with their account on the Manuscript Tracking System prior to acceptance. ORCID helps the scientific community achieve unambiguous attribution of all scholarly contributions. You can create and link your ORCID from the home page of the Manuscript Tracking System by clicking on 'Modify my Springer Nature account' and following the instructions in the link below. Please also inform all co-authors that they can add their ORCIDs to their accounts and that they must do so prior to acceptance.

Version 1:

Decision Letter:

Dear Professor de Vries,

Your manuscript titled "Assessing Economic Benefits of International Cooperation over Marine Plastic Pollution" has now been seen by our reviewers, whose comments appear below. In light of their advice we are delighted to say that we are happy, in principle, to publish a suitably revised version in Communications Earth & Environment.

We therefore invite you to edit your manuscript to comply with our format requirements and to maximise the accessibility and therefore the impact of your work.

EDITORIAL REQUESTS:

****Please take care to match our formatting and policy requirements. We will check revised manuscript and return manuscripts that do not comply. Such requests will lead to delays. ****

SUBMISSION INFORMATION:

OPEN ACCESS:

Communications Earth & Environment is a fully open access journal. Articles are made freely accessible on publication. For further information about article processing charges, open access funding, and advice and support from Nature Research, please visit <https://www.nature.com/commsenv/open-access>

Link Redacted

Best regards,

Yuwan Duan, PhD
Editorial Board Member
Communications Earth & Environment

Martina Grecequet, PhD
Senior Editor,
Communications Earth & Environment
@CommsEarth

REVIEWERS' COMMENTS:

Reviewer #1 (Remarks to the Author):

Authors have revised this paper according to my comments. I have no further comments

Reviewer #2 (Remarks to the Author):

Thank you for sharing the revised manuscript, **"Assessing Economic Benefits of International Cooperation over Marine Plastic Pollution"** by Professor de Vries and colleagues.

Having reviewed the updated version, I am pleased to confirm that my comments have been fully addressed. The revisions have significantly improved the manuscript, and I believe it is now suitable for acceptance.

Thank you for allowing me to contribute to the review process, and I look forward to seeing the published work.

Reviewer:

Assessing Economic Benefits of International Cooperation over Marine Plastic Pollution

Summary of the Paper:

The paper estimates the economic consequences of international cooperation in controlling marine plastic pollution (MPP) by analyzing the experiences of 16 North Atlantic countries. It develops a framework combining plastic transfer coefficients with a game-theoretic model to evaluate the potential economic gains of cooperation. The paper highlights how coordinated efforts can lead to significant MPP reductions and increased economic benefits, contrasting fully cooperative and non-cooperative scenarios. It explores political economy considerations, which are crucial for achieving equitable outcomes among nations and understanding the constraints that influence cooperation.

Strengths of the Paper:

1. Integration of oceanographic modeling with Game Theory:

The paper stands out by merging oceanographic data on plastic transfer dynamics with game-theoretic models. This combination enhances the understanding of how marine plastic pollution crosses national boundaries and provides a quantitative basis for analyzing economic efficiency in international cooperation. The use of plastic transfer coefficients is a novel approach that incorporates environmental factors into the economic analysis, making the results more relevant for policymakers.

2. In-depth analysis of cooperation scenarios:

By comparing non-cooperation with full cooperation, the paper offers a thorough and well-structured economic analysis. It provides insights into how varying levels of international cooperation impact pollution reduction and welfare outcomes. The addition of political economy constraints, such as equal abatement and equal benefits, deepens the analysis by reflecting real-world complexities in international agreements.

3. Policy-relevant findings:

The paper's findings have practical relevance for international policymaking. It quantifies the economic benefits of cooperation on MPP, offering policymakers evidence to support the formation of cooperative agreements. It emphasizes the need for region-specific solutions and demonstrates that substantial gains can be achieved through joint action, which is particularly relevant for ongoing negotiations under the UN framework for a global plastic treaty.

Key Claims:

1. Economic benefits of cooperation:

The authors convincingly argue that international cooperation leads to significant economic benefits, backed by a model that integrates plastic transfer dynamics with willingness-to-pay data. The application of game theory to MPP is innovative and provides valuable insights into cooperative strategies.

2. Unequal distribution of benefits:

The paper highlights that the benefits of cooperation are unevenly distributed, with wealthier nations, such as the United States and Germany, gaining more from MPP reduction efforts while large burdens of plastic abatement falling on a few, low-income countries. This conclusion is supported by economic modeling that considers income disparities, plastic emissions, and transfer coefficients.

Suggestions for improvement:

1. Incorporate more empirical data on the costs of MPP in sectors like tourism and fisheries to broaden the context.
2. Expand the analysis to include different types of plastics, beyond macroplastics.
3. Add dynamic elements to the model to account for changing plastic emission trends over time.

Conclusion:

The paper makes a valuable contribution to environmental economics by applying game theory to international cooperation on marine pollution. Limitations of the study should be clarified. Nevertheless, the work provides a strong foundation for understanding the economic benefits of cooperative MPP management.

Responses to the Reviewers

COMMSENV-24-2063-T

Assessing Economic Benefits of International Cooperation over Marine Plastic Pollution

We are grateful to the two reviewers for their valuable and constructive comments on our paper. In our revision we have made numerous changes in response to your suggestions, which we believe have significantly improved the manuscript. In this response we have reproduced the main comments, followed by our responses highlighted in *italics*.

Comments Reviewer #1 (Remarks to the Author):

Our responses are in italic text.

This study assesses the economic benefits of international cooperation in addressing marine plastic pollution, a critical global issue. I hope my comments will contribute to the improvement of this valuable research.

RESPONSE: We greatly appreciate the positive response of the reviewer to our paper and constructive suggestions for improvements. We hope we have been able to improve the paper in responding to them, as explained further in our responses below.

The primary objective of this study is to demonstrate that international cooperation is essential for tackling the problem of plastic pollution. However, I believe the study should also address why international cooperation is often challenging. What are the key factors that hinder effective international collaboration on this issue? Additionally, what strategies can be employed to overcome these challenges?

RESPONSE: We thank the reviewer for this suggestion and agree. We have added a new paragraph to the manuscript on page 2 (beginning on line 20) to better explain why international cooperation is both necessary and challenging for marine plastics, along with strategies to address these challenges:

“International cooperation to address reductions in any transboundary global or regional pollutant is challenging, because the benefits of these reductions have the characteristics of a public good.¹⁶ The physical movement of plastic in and across international waters implies that some of the environmental damages due to plastic waste emitted from one country may be imposed on other countries. Emissions reductions by any one country (e.g., country A) also decrease marine plastics in the waters and beaches of other countries (e.g., countries, B, C and D), even if the latter countries make no efforts to reduce their own emissions. Consequently, although expenditures on MPP reductions from any one country are incurred by that country alone, the benefits of unilateral abatement actions are experienced by all countries whose territorial waters and shorelines are impacted. The resulting non-excludability of benefits (i.e., actions by one country necessarily benefit others) creates a strategic cooperation problem, since it can be in each country’s selfish best interest to rely on abatement actions undertaken by other countries with which they share the ocean resource, rather than reducing emissions themselves. Possible strategies to solve this cooperation dilemma include (i) international agreements whereby each country agrees to a legally-binding mandate to reduce its own emissions if other countries also act, and/or (ii) cooperative agreements wherein countries offer each other side payments to induce greater cooperation.^{14,15”}

The authors estimate the plastic transfer matrix using a game theory model. I find it difficult to understand how such a transfer matrix can be accurately estimated based on a game-theoretical approach. The movement and distribution of plastic pollution are influenced by numerous physical factors. How does the study justify estimating this matrix using an economic model rather than a model grounded in physical science?

*RESPONSE: We regret that the original paper was not clearer about the source of the plastics transfer matrix. The transfer matrix is **not** based on a game-theoretical model—a point that we now make more explicitly clear in the revised manuscript. Instead, the transfer matrix is an output of a physical particle tracking model developed for this study by one of the named authors.*

We have made multiple revisions to clarify this issue. First, we have revised the abstract of the paper to clarify the source of the transfer matrix: “The framework integrates an estimated plastic transfer matrix from a particle tracking model with game theory to derive the potential economic benefits of international cooperation for 16 countries bordering the North Atlantic Ocean.”

This particle tracking model itself is described on page 4 (beginning on line 81) in the revised manuscript (with extensive details beginning on line 354), building on text from the original paper:

“The MPP transfer coefficients were calculated using a particle tracking model²⁶ with gridded surface ocean currents and wind data^{27,28} which was constructed specifically for this project; and annual estimates of river plastic emissions⁸. We simplify the model by fixing annual emissions. This step simplifies the economic modelling by providing a set of transfer coefficients that are in near steady state, subject to remaining interannual variations due to variable ocean and wind forcing. A parameterisation of plastic removal was used to account for the selective loss of buoyant plastic from surface waters to the subsurface and seafloor¹⁰, and to bring the model to an approximate steady state. This was calibrated to align estimates of plastic river inflows with the generally smaller estimates of surface ocean plastic inventories that have been derived from observational and model-led studies^{20,29}. The idealised setup accounts for different plastic residence times within EEZs and solves the problem of lag effects associated with varying plastic transit times between countries, with the caveat that time-varying inputs are not accounted for. The final model yields a matrix of transfer coefficients, where each value gives the annual average fractional contribution of each country’s emissions to the stock of plastic in each country’s EEZ (Figure 1). Details about the model assumptions, its calibration and estimation of these figures are given in Methods and in the Supplementary Information (SI).”

This particle tracking model is used to generate the transfer coefficient matrix, which then becomes an important component of the optimisation model, since it determines how the flow of plastic emissions from country X impact on the national waters of all other countries in the model.

The study relies heavily on Maler’s work, highlighting the differences between Marine Plastic Pollution (MPP) and sulphur dioxide (SO₂) pollution. However, are there other models more suitable for analyzing MPP, given its unique characteristics and the high degree of uncertainty associated with it? The authors should clarify why Maler’s model was chosen and consider whether alternative models might be more appropriate for this study.

RESPONSE: We thank the reviewer for this constructive suggestion. The revised paper now includes additional text to explain how the main features of the ocean plastics control problem map onto Maler’s framework, and that an adaptation of this framework (as developed here) is an ideal approach for analysing the marine plastics coordination problem. This revised text is found in the Introduction on page 3 (beginning on line 40):

“We integrate these elements using an adaptation of the game-theoretic framework in Mäler¹⁷, initially applied to the case of sulphur dioxide (SO₂) emissions that contribute to acid rain¹⁸. The features of the control problem which make our adaptation of Mäler’s approach particularly appropriate are (i) representation of the physical realities of pollutant transfer across the North Atlantic to different countries, (ii) variation in both the valuation of damage costs across the affected countries, along with variation in the costs of reducing pollution, (iii) an optimisation framework in which the benefits of cooperation can be quantified relative to a baseline of no cooperation, and (iv) each country’s incentive to behave non-cooperatively in terms of abatement expenditures. Results from the integrated assessment provide insight into key questions for international coordination, including: (1) What abatement policy maximises the net economic benefits of international cooperation in MPP reduction? (2) How are the benefits of international cooperation distributed between different countries under the optimal cooperative solution? (3) What is the impact of political economy constraints on these cooperative outcomes, in terms of both the overall benefits of cooperation, and the reduction in MPP?”

If the reviewer has a particular alternative approach in mind and would like us to comment on this approach (e.g., with regard to advantages/disadvantages relative to the presented model), we can add something along those lines to the paper. However, we are unaware of a realistic alternative modelling approach which incorporates all of these main features of the control problem and would hence provide a more appropriate alternative. This is the reason that an approach based on Mäler’s seminal work was chosen.

Figure 4: The authors mention that the scenario of equal value is not depicted. However, it appears that an equal value scenario is present in Figure 4b. Could the authors please clarify this discrepancy?

RESPONSE: Thank you for highlighting this point, and we’re sorry this has caused some confusion. Figure 4 depicts results from the economic optimisation model for the year 2014 under the four alternative political-economy constraints (i.e., C1, C2, C3 and C4) that we consider in the paper. In Figure 4a, under the “equal value” constraint (C2) the value was the same for all countries (i.e., 0%). In Figure 4b, under the “equal abatement” constraint (C3) abatement was the same for all countries (i.e., 43%). Since there is no variation across countries, we decided not to include these straight lines in the relevant figures and just report the values in the text under Figure 4. All the associated quantitative data is included in Extended data Table 3.

Comments Reviewer #2 (Remarks to the Author):

Our responses are in italic text.

Conclusion:

The paper makes a valuable contribution to environmental economics by applying game theory to international cooperation on marine pollution.

RESPONSE: We greatly appreciate the positive response of the reviewer to our paper and constructive suggestions for improvements. We hope we have been able to improve the paper in responding to these comments. Revisions that have been made to the paper are described below, under each of the reviewer's suggestions.

Limitations of the study should be clarified.

RESPONSE: We thank the reviewer for this constructive suggestion. We have greatly expanded the Discussion section of the revised paper to more thoroughly explain the assumptions and limitations of the presented approach. We emphasize that assumptions and limitations such as these are common in large-scale integrated models and unavoidable in some form to maintain tractability — however we agree fully on the importance of making them clear. This new discussion is integrated with a corresponding discussion of possible extensions to the model discussed in the reviewer's second main point below ("suggestions for improvement"), so we not only discuss current limitations but also possible approaches that could be used to address these limitations in the future, with the addition of further data and models beyond those currently available. These revisions are found throughout the entirety of the new Discussions section, beginning on line 234 in the revised manuscript.

Nevertheless, the work provides a strong foundation for understanding the economic benefits of cooperative MPP management.

Suggestions for improvement:

RESPONSE: The suggestions that follow all address possible extensions to the model that might, in principle, be explored. We have made multiple revisions to the manuscript to address this topic. First, we now present a detailed and much expanded section in the Discussion that considers extensions to the model, including sensitivity analyses that are possible (and have been conducted) along with additional, future extensions that could be feasible with considerable new data and potential development of novel model structures. Although the latter extensions are not possible without additional funding and several years of effort (beyond what we now present in the paper and appendices), we can nonetheless make useful qualitative statements (based on the model structure and properties) about how the results are likely to change.

Among future extensions that could be possible with new data, we now describe (in the Discussions section) a range of possibilities, how they could be implemented, and the new data or approaches that would be necessary to do so. These include possible extensions such as (i) accommodation of possible changes in plastic emission trends over time and the corresponding lack of a steady state, (ii) inclusion of additional types of plastic within the model, and (iii) consideration of additional plastic damages. Further details on each of these revisions is described in the responses below.

1. Incorporate more empirical data on the costs of MPP in sectors like tourism and fisheries to broaden the context.

RESPONSE: We have added new text to the manuscript outlining how future work could add to the comprehensiveness of our current estimates of damage costs – noting that the damage costs we use already likely include impacts on tourism (via the attribute of plastics pollution on beaches and in coastal waters of each country), and aspects of impacts on fisheries (via the attribute of plastics pollution in coastal and international waters). Our respondents were made aware of this wide range of impacts in the survey instrument used to collect the stated preference data, and their willingness to pay already includes, therefore, at least some of the tourism and fisheries impacts mentioned in this comment. Hence, it would not be valid to add new, stand-alone estimates of tourism and fisheries damages to the existing damage costs in the model, as this would cause at least some degree of double counting.

See Börger et al. (2023) for full details of these survey instruments along with the “choice modelling information” in the SI of this paper:

Börger, T., Hanley, N., Johnston, R.J., Meginnis, K., Ndebele, T., Ali Siyal, G.E., de Vries, F. (2023), “Equity Preferences and Abatement Cost Sharing in International Environmental Agreements,” American Journal of Agricultural Economics 1-26. <https://doi.org/10.1111/ajae.12392>

The revised paper also notes that (beginning on line 254):

“Further steps to improve the presented framework could include incorporation of (i) additional abatement benefits, (ii) different types of plastic, (iii) a more expansive set of countries, and (iv) original data on abatement costs. For example, our results may understate the full economic benefits of reducing MPP, since the valuation data used to estimate the benefits of reduced MPP for each country relate only to the values citizens attach to reducing pollution in their own national beaches and coastal waters³⁰. But citizens may also value MPP reductions in international waters and in third-party countries, implying that the benefits of abatement could be higher. Future work could expand the framework to incorporate broader representation of the environmental costs of MPP and hence the benefits of abatement.⁴¹ Moreover, as emphasized above, the approach (and WTP estimates in Börger et al.³⁰) only consider reductions in buoyant macroplastic, whereas an IEA could also reduce stocks of other types of plastic.”

In summary, we agree that more comprehensive damage costs would be useful and explain this issue in the revised paper. We have added a discussion of this topic to the revised paper, while acknowledging that adding these costs empirically (and in a valid manner) would require new data and methods beyond those currently available.

2. Expand the analysis to include different types of plastics, beyond macroplastics.

RESPONSE: This is an excellent suggestion in principle but addressing it in full would require (essentially) a new study and corresponding data to address the unique properties and damages of these different types of plastics. Going beyond macroplastics to, for example, nano-, micro-, meso- and mega- plastics is a logical next step. However, each of these types of plastic has unique transport properties and damages that cannot be incorporated into the framework without new data and modelling. At the current time, there is limited, standardised data on all these types of plastic and their associated emissions to the marine environment; and

adequate data on households' willingness to pay to reduce levels of these different types of plastic is not available (the Börger et al. (2023) study only addresses the type of macroplastic considered in the present framework). As such, an expansion to fully cover different size-classes of plastic is far beyond the scope of the current study.

However, it is possible to explore what impact changing the type of plastic would have, by recognising that different types of plastic can be characterised by parameters that govern their transport properties. Following the reviewer's suggestion, we have added this new material to the revised paper as part of our sensitivity and robustness analysis. For buoyant plastic, a key property is their buoyancy in sea water, which is a function of their density and the type of polymer from which they are constructed. Less buoyant plastics tend to sit lower in the water, with a smaller portion of their surface area exposed to the wind. This acts to reduce the importance of the wind in governing their transport dynamics, which in the model is reflected in a lower wind drift factor. For objects that are fully submersed, there is no direct wind impact; although the wind can still influence the movement of objects indirectly, through its driving of surface ocean currents and mixing. In the original submission, we included a sensitivity study into the impact of varying the wind drift factor on particle transport dynamics (Extended Data Figure 5). Due to the computational cost of running the particle tracking model, we did this for a single year of emissions and calculated the impact using a metric for exposure (defined in the SI). However, these results were not explicitly discussed in the main paper.

To help address this point, we have now included a new paragraph in the main article in which we explore how different types of plastic move, and the impact this has on the exposure of countries to the emissions of all other countries represented in the study. The sensitivity study we conducted already discusses the robustness of our findings to changes in the transport coefficients.

The revised paper also clarifies why the model is purposefully limited to macroplastics: "Because the ocean transport and environmental impacts of marine plastics vary over different types of plastic (e.g., micro- versus macroplastics)⁴, the information required to implement the model, and the potential answers to our research questions, also differ. We thus limit the scope of the study by focussing on floating macroplastic (>0.5 cm in size) that has entered the marine environment via rivers, before being moved within and between each country's territorial waters."

3. Add dynamic elements to the model to account for changing plastic emission trends over time.

RESPONSE: As above, this is an excellent extension to consider for the future but is infeasible in the context of the current effort. However, we can make useful qualitative statements (based on the model structure and properties) about how results would be likely to change, were changing plastic emission trends somehow able to be incorporated.

More specifically, on the direct incorporation of changing trends over time into the model, we note that we would need data on projected future emissions and income to infer changing cost parameters over time. We could then estimate the cooperative and Nash solutions in each time period and report differences of discounted present value of each path (cooperative vs Nash). However, this would require the development of new and more complex optimisation techniques to accommodate an infinite time horizon in the model. A truly comprehensive approach would require development of a full-scale integrated assessment model (IAM) to incorporate feedback between abatement policies and the evolution of abatement technology and income growth. That would be a massive research task and would require still additional assumptions to maintain tractability (and would be potentially sensitive to assumed model structures, etc.). Hence, it would be akin to a new research project. At the same time, we can speculate qualitatively (based on the current model structure) how the results might change were such an approach to be developed.

This topic (along with those mentioned above) is now addressed explicitly in a new paragraph we have added to the revised manuscript, at the beginning of the Discussion section on page 8 (beginning on line 235):

“Our research develops a generalisable framework for analysing the potential gains from international cooperation when reducing MPP. The analysis demonstrates how ocean modelling can be linked to economic valuation and game-theoretic analysis to assess the benefits of cooperation, and the effects of alternative political-economy considerations on the IEA performance. Results were drawn from a particular case study — buoyant macroplastic pollution in the North Atlantic. At present, detailed knowledge regarding key model components remains uncertain, including knowledge regarding the movement of plastic between different reservoirs as well the benefits and costs of MPP abatement beyond the scenarios included in our stated preference study, which includes buoyant macroplastic only, and only samples residents in two of the North Atlantic countries (the US and the UK). The analysis was able to assess the robustness of the findings regarding a number of assumptions. However, future studies could explore potential extensions of the approach to incorporate features such as a more comprehensive representation of how plastic moves from sources to sinks⁴⁰ (while accounting for associated uncertainties), time-varying conditions (e.g., MPP inputs; changes in GNI), or potential feedbacks between the introduction of abatement targets and enhanced abatement technologies. For example, we assume that MPP has reached a steady-state equilibrium in the physical environment. But MPP has been steadily increasing over time, is forecast to continue increasing, and moves slowly in the ocean—features that reduce the likelihood of a steady state. Moreover, if MPP is increasing over time, gains from international cooperation are likely to be higher than those reported here. Such effects could be explored using an Integrated Assessment Model (IAM), building on the framework developed here.”